# A High-Fat Diet Induces Epigenetic 1-Carbon Metabolism, Homocystinuria, and Renal-Dependent HFpEF

**DOI:** 10.3390/nu17020216

**Published:** 2025-01-08

**Authors:** Suresh C. Tyagi

**Affiliations:** Department of Physiology, University of Louisville School of Medicine, Louisville, KY 40202, USA; suresh.tyagi@louisville.edu

**Keywords:** folate 1-carbon metabolism, heart failure, ATP–citrate lyase, gene writer, eraser and editor, kidney, morning heart attacks

## Abstract

Background/Objectives: Chronic gut dysbiosis due to a high-fat diet (HFD) instigates cardiac remodeling and heart failure with preserved ejection fraction (HFpEF), in particular, kidney/volume-dependent HFpEF. Studies report that although mitochondrial ATP citrate lyase (ACLY) supports cardiac function, it decreases more in human HFpEF than HFrEF. Interestingly, ACLY synthesizes lipids and creates hyperlipidemia. Epigenetically, ACLY acetylates histone. The mechanism(s) are largely unknown. Methods/Results: One hypothesis is that an HFD induces epigenetic folate 1-carbon metabolism (FOCM) and homocystinuria. This abrogates dipping in sleep-time blood pressure and causes hypertension and morning heart attacks. We observed that probiotics/lactobacillus utilize fat/lipids post-biotically, increasing mitochondrial bioenergetics and attenuating HFpEF. We suggest novel and paradigm-shift epigenetic mitochondrial sulfur trans-sulfuration pathways that selectively target HFD-induced HFpEF. Previous studies from our laboratory, using a single-cell analysis, revealed an increase in the transporter (SLC25A) of s-adenosine–methionine (SAM) during elevated levels of homocysteine (Hcy, i.e., homocystinuria, HHcy), a consequence of impaired epigenetic recycling of Hcy back to methionine due to an increase in the FOCM methylation of H3K4, K9, H4K20, and gene writer (DNMT) and decrease in eraser (TET/FTO). Hcy is transported to mitochondria by SLC7A for clearance via sulfur metabolomic trans-sulfuration by 3-mercaptopyruvate sulfur transferase (3MST). Conclusions: We conclude that gut dysbiosis due to HFD disrupts rhythmic epigenetic memory via FOCM and increases in DNMT1 and creates homocystinuria, leading to a decrease in mitochondrial trans-sulfuration and bioenergetics. The treatment with lactobacillus metabolites fat/lipids post-biotically and bi-directionally produces folic acid and lactone–ketone body that mitigates the HFD-induced mitochondrial remodeling and HFpEF.

## 1. Introduction

Chronic gut symbiosis due to a high-fat diet (HFD) instigates cardiac remodeling and heart failure with preserved ejection fraction (HFpEF) [1,2,3,4]; however, its mechanisms are largely unknown. The therapeutic approaches focusing only on the oxidative stress, inflammation and hyperlipidemia have not yielded significant positive clinical outcomes for HFpEF. Although HFD contains lipids, fats, and proteins, there is not enough information regarding the role of animal protein intoxication in HFpEF. Interestingly, the recent studies report that the ATP citrate lyase (ACLY) supports cardiac function and decreases more in HFpEF than HFrEF. Paradoxically, ACLY creates hyperlipidemia. Because the inhibition of ACLY (ACLYi) mitigates hyperlipidemia and fatty acid declines ACLY expression [5], the connection is unclear. Epigenetically, ACLY acetylates histone [6]. It is known that the transcription is controlled by on/off promoters by rhythmic methylation/de-methylation during development, health and disease. The chromatin maturation, adaption, and accessibility are regulated by acetylation. The epigenetic folate 1-carbon metabolism (FOCM) pathways recycle Hcy back to methionine. The DNA methylation by FOCM [1,2,3,4,7] is the hallmark of epigenetic memory, i.e., rhythmic gene imprinting and off-printing during embryogenesis, development, health, remodeling, and diseases [8]. One hypothesis is that an HFD induces epigenetic FOCM and HFpEF. Probiotics utilize fat/lipids [9,10] which post-biotically increase mitochondrial bioenergetics and attenuate HFD-induced heart failure. Here, we review a novel and paradigm-shift epigenetic mitochondrial sulfur trans-sulfuration pathways that selectively targets HFD-induced HFpEF. A HFD via transport and metabolism pathways instigates cardiac remodeling.

## 2. The Role of Kidney in HFpEF

Although the pathologies of obesity, hypertension, aging and diabetes are associated with kidney dysfunction, the role of kidney in HFpEF is not clear. Interestingly, we have shown the contribution of renal de-nervation in HFpEF [11]. The HFpEF is afterload and renal-dependent diastolic dysfunction; the cardiac relaxation rate, i.e., -dp/dt/MAP), normalized with MAP, is an effective and more appropriate test to determine levels of HFpEF. On the other hand, the HFrEF is pre-load dependent, leading to systolic dysfunction [12]; therefore, LV pressure should be used to normalize LV weight, in order to determine the levels of HFrEF (Figure 1). These studies will be a better index of HFpEF. To this end, it is important to address the role of epigenetics in HFpEF.

## 3. Epigenetics of HFpEF

Methionine (Met) is a substrate for epigenetic DNA/RNA/proteins, histone methylation, and the generation of homocysteine (Hcy) via SAM/SAH/SAHH pathways. Hcy are then converted to H2S via trans-sulfuration 3MST/CBS/CSE pathways [13,14,15]. The inhibition of methyl transferase restores metabolism and improves the regenerative capacity of the muscle [16]. The mutation in DNMT causes hypermethylation and causes growth retardation [17]. The accumulation of by-product uric acid causes HFrEF [18]. The activation of growth arrest DNA damage (GADD) is associated with the activation of nuclear metalloproteinases [19] (Figure 2). Collectively, these studies suggest the role of epigenetics in intra-nuclear, chromatin, histones, and extracellular remodeling.

Although a chronic high-fat diet (HFD) is associated with hypertension/obesity and “double hit” heart failure [20,21,22], its mechanism is unclear. There is ~1% of methionine (Met) in HFD [23], which is significantly high for a single amino acid. Interestingly, studies from our laboratory [1,2,3,4] observed an increase in dysbiosis due to an HFD (i.e., increase in ratio between polyunsaturated fatty acid/monounsaturated fatty acid in WT mice+HFD). The treatment with lactobacillus-probiotic (PB, a folate and lactone–ketone body producer [24,25,26] mitigated this increase in poly-unsaturated fatty acid [1,2,3,4]. Due to high methionine, gut–dysbiosis instigates a de-arrangement in the epigenetic rhythmic methylation/demethylation ratio on the DNA/RNA, and protein/histones, due to a DNMT1/TET2 ratio [27], and consequently, creates hyperhomocysteinemia (HHcy) [1,2,3,4,8]. Previous studies from our laboratory, using a single-cell analysis, revealed an increase in SLC25A (transporter of s-adenosine-methionine (SAM)) during elevated levels of homocysteine (Hcy) [28], caused by impaired recycling of homocysteine back to methionine via an increase in epigenetic methylation on H3K4, K9, and H4K20) via folate 1-carbon metabolism (FOCM) pathway by gene writer (DNMT) and decrease in eraser (TET/FTO) (Figure 2).

Although clinical trial shows that microbiome metabolizes cholesterol/lipids [9,10] and lactobacillus elicits cardiac benefits [9], it is unclear whether the treatment with *Lactobacillus rhamnosus,* a ketone body fuel for mitochondria [25,26,29,30,31,32,33,34] and a folic acid producing probiotic [24,25,26], reverses dysbiosis-induced cardiac complications, in part by increasing mitochondrial trans-sulfuration, H_2_S, and bioenergetics [26] (Figure 3). Therefore, it is important to determine, for the first time, that a probiotic can increase mitochondrial sulfur metabolism, trans-sulfuration and bioenergetics and mitigate HFD-induced obesity and heart failure.

Gut dysbiosis, FOCM, methylation, and HHcy via DNMT, BHMT, and PEMT are linked to microbiomes (Figure 4). We suggest an increase in DNMT and PEMT and decrease in BHMT in HFD-fed mice, which will increase Hcy [1,2,3,4]. The treatment with probiotics decreased levels of DNMT and PEMT; however, there were no changes in BHMT [1,2,3,4]. These results suggest differential mechanisms, i.e., betaine versus folate pathways of re-methylation of Hcy [33,34,35] (Figure 4). Interestingly, this also suggested a prominent central role of DNMT1 in Hcy formation by HFD. This incited us to employ DNMT1 knockout mice. We observed that gut–dysbiosis and altered Hcy metabolism represent the dominant mechanism wherein HFD induces DNMT1, causing detrimental cardiac remodeling and mitochondrial metabolic dysfunction (i.e., trans-sulfuration and bioenergetics) during HFD-induced HFpEF.

Others [7,36] have also supported our hypothesis that there is a rhythmic methylation/demethylation during the mitochondrial TCA cycle by epigenetic gene writer (DNMT) and erasers (TET and FTO). The premise of this study is also supported by Kay et al. [37,38], and Engler et al. [39,40,41], who reveal the role of transforming fibroblasts, and fibrosis in cardiac remodeling. The study by Dees et al. [42] suggests that DNA-methyltransferase 3A (DNMT3A) and DNMT1 in fibroblasts silence the expression of suppressor of cytokine signaling 3 (SOCS3) in a SMAD-dependent manner by promoting hypermethylation. The downregulation of SOCS3 facilitated the activation of signal transducers, and activators of transcription 3 (STAT3) to promote fibroblast-to-myofibroblast transition, collagen release, and fibrosis in vitro and in vivo. The re-establishment of the epigenetic control of STAT3 signaling via the genetic or pharmacological inactivation of DNMT3A reversed the phenotype of fibroblasts in tissue culture, inhibited TGFb-dependent fibroblast activation, and ameliorated experimental fibrosis in murine models [42,43,44]. Felisbino and McKinsey suggested the role of epigenetics in cardiac fibrosis with emphasis on inflammation and fibroblast activation [45,46]. The authors demonstrated the role of histone acetylation, and reader proteins using a small-molecule inhibitor of bromo-domain-containing protein 4 (BRD4) for mitigating cardiac fibrosis [47]. Interestingly, along the same line, we demonstrated that Hcy induces collagen expression in a dose- and time-dependent manner, as measured by Northern blot analysis [48]. Hcy causes fibroblast activation, and myofibroblast differentiation in murine aortic endothelial cells [49]. Previously, in human, we show the role of a dis-integrin and metalloproteinase (ADAM) in connexin-43 (Cx43) degradation in human end stage heart failure [50]. Others have shown that Cx-43 synchronizes myocyte-mitochondrial function [51,52]. Here, we focus on the significance of epigenetics via gene writer, eraser, and reader functions along with direct consequences of Hcy in cardiac remodeling.

## 4. Homocysteine and Lipid Connection

Most of the human population is mildly HHcy and asymptomatic; however, when on an HFD, especially a red-meat-rich diet that is high in methionine (a substrate for Hcy generation during epigenetically regulated DNA methylation) [1,2,3,4], severe HHcy could develop, and that will ultimately cause symptomatic cardiovascular diseases. Interestingly, although folic acid and H_2_S reduce obesity and brown adipose fat [14,15], their mechanisms are unclear.

Homocysteine theory suggests a diet high in animal protein and low in B vitamins, which occur in many foods but are very easily destroyed by processing, i.e., a diet of meat, cheese, milk, white flour, and foods that are canned, boxed, refined, processed, or preserved. The theory suggests a strong connection between diet and CVD, but one that is different from cholesterol/lipids. The homocysteine theory considers CVD due to what McCully calls protein intoxication [53,54]. The cholesterol theory (sometimes called the lipid theory) instead demonizes fats. Since proteins and fats often occur in the same foods, the potential dietary treatments for high homocysteine and high cholesterol are similar, with the following distinction: the anti-homocysteine diet focuses on what should be eaten, as a preventive, while the anti-cholesterol diet focuses on what should be avoided, as a precipitator. Thus, a diet of lower homocysteine includes many natural sources of B vitamins, like fresh fruits and vegetables, and limits animal protein. A cholesterol-reducing diet limits foods high in saturated fats and cholesterol, like eggs, meat, and butter. Unfortunately, the latter is more commercially popular. Therefore, we use lactobacillus, which produces folic acid and lactone–ketone bodies (food for mitochondria) post-biotically and mitigates HFpEF.

The probiotic, post-biotically produced folic acid and lactone–ketone bodies mitigate HFD-induced heart failure and are therapeutically innovative (i.e., killing two pathologies in one strike). Interestingly, some probiotics also lower cholesterol [10]; however, their mechanisms are unclear. In this novel review article, we address the fact that lactobacillus eats lipids and post-biotically produces folic acid and lactone–ketone bodies, a bi-directional probiotic that mitigates HFD-induced mitochondrial remodeling and heart failure.

## 5. Viewpoint on the Role of Epigenetics and Future Perspective

This review presents an innovative hypothesis that the therapeutic effect of probiotic treatment should be tested to reverse gut–dysbiosis effects in HFpEF. Describing the mechanisms through which diet (particularly an HFD) and dysbiotic events in distant organs, such as the intestine, affect heart health is important, and positive findings will support the relevance and impact of systemic health on heart health. Although the use of probiotics as a therapeutic alternative to heart disease has been previously studied in pre-clinical models and clinical trials, there is no mechanism-based simple and safe therapy using probiotics. Here, we suggest that a probiotic that produces folic acid (a Hcy lowering agent) and lactate (a ketone body, fuel for mitochondria) post-biotically can eat fat and increases mitochondrial bioenergetics.

Based on this review, it is important to determine whether the lactobacillus downregulates SLC25A, and epigenetic gene writer DNMT1 upregulates eraser TET2/FTO and attenuates cardiac dysfunction by lowering homocysteine. The lactobacillus post biotically produces folic acid and attenuates HHcy via a decrease in epigenetic gene writer (DNMT1) and increase in eraser (TET2/FTO). Also, it will be novel to determine whether the lactobacillus increases SLC7A and mitochondrial H_2_S production by trans-sulfuration. Lactobacillus post-biotically produces a lactone/ketone body and improves mitochondrial bioenergetics via trans-sulfuration (Figure 5). The use of DNMT1KO and mitigation of HFD-induced epigenetic de-arrangement is novel. The use of transgenic mice overexpressing TET2, FTO (erasers), CBS (transsulfuration), and TFAM (mitochondrial) is mechanistically innovative.

## Figures and Tables

**Figure 1 nutrients-17-00216-f001:**
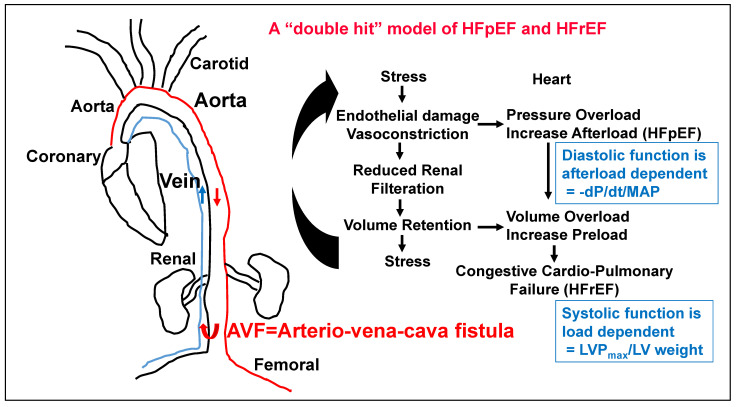
The chronic volume overload by aorta–vena cava fistula (AVF) creates HFpEF and HFrEF.

**Figure 2 nutrients-17-00216-f002:**
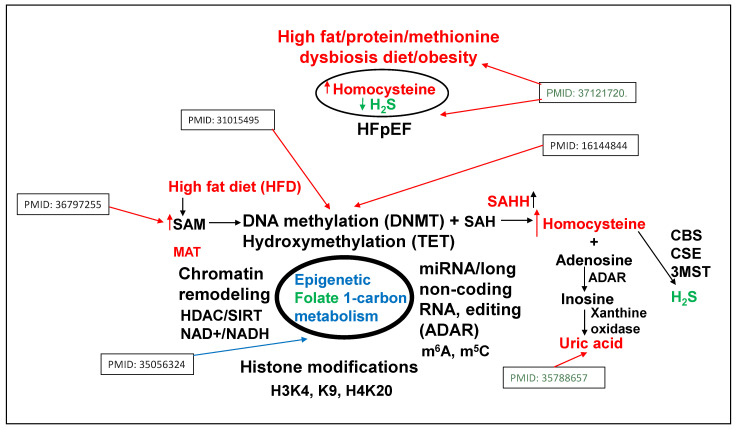
Metabolic regulation of epigenetic memory via FOCM by gene writer (DNMT), gene eraser (TET), and homocysteine in HFD-induced cardiac dysfunction [13,15,16,17,18,19].

**Figure 3 nutrients-17-00216-f003:**
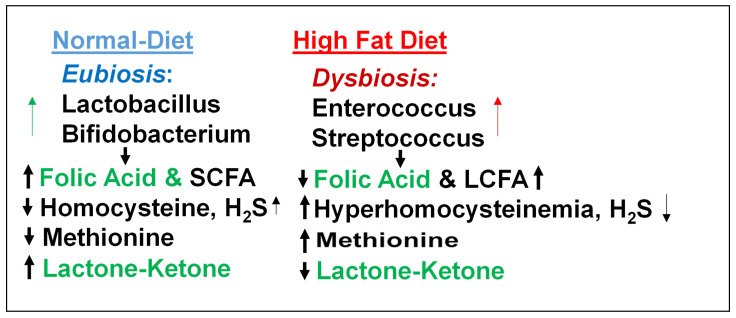
Normal eubiosis contains beneficial microbiomes, lactobacillus, and SCFA, whereas dysbiosis contains enterococcus and LCFA. Therefore, it is important to increase eubiosis via lactobacillus.

**Figure 4 nutrients-17-00216-f004:**
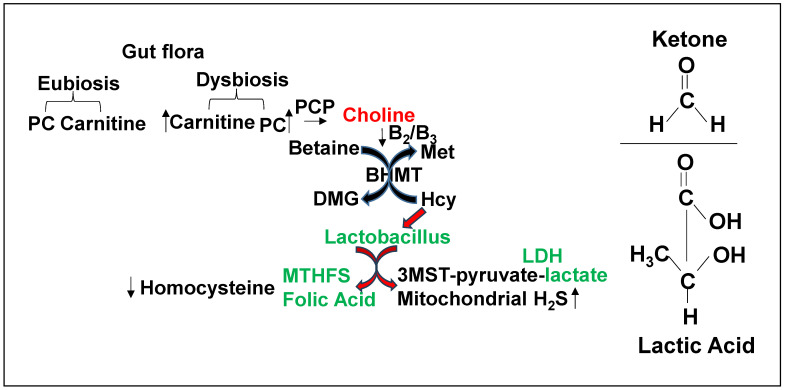
A link between gut–dysbiosis and a decrease in betaine homocysteine methyltransferase (BHMT)-dependent re-methylation of homocysteine is elicited. The lactobacillus produces folate and ketone/lactate, post-biotically, and therefore decreases Hcy and increases mitochondrial bioenergetics, because lactate/ketone fuel for mitochondria increases H2S and mitigates HFD-induced cardiac dysfunction.

**Figure 5 nutrients-17-00216-f005:**
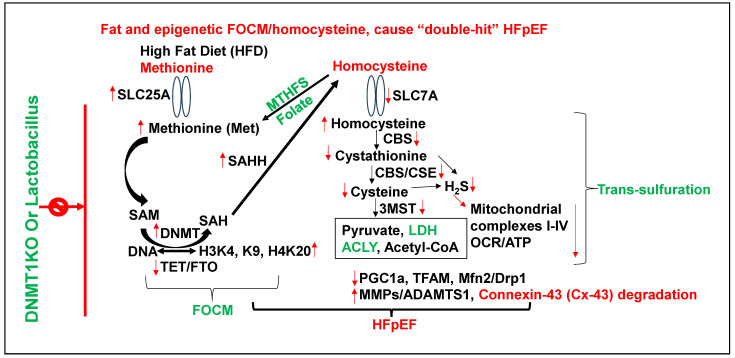
The hypothesis is that dysbiotic HFD increases transporter SLC25A and rhythmic bodies, i.e., methylation of histone lysine by gene writer and eraser, creating hyperhomocysteinemia (HHcy). The transporter SLC7A is decreased, causing disruption in mitochondrial sulfur metabolism H_2_S. The probiotic lactobacillus post-biotically produces folic acid and a lactone–ketone body, converting Hcy back to methionine and lactone (fuel for mitochondria), respectively, improving mitochondrial bioenergetics.

## Data Availability

Data are contained within the article.

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
