# Peer review of "A High-Fat Diet Induces Epigenetic 1-Carbon Metabolism, Homocystinuria, and Renal-Dependent HFpEF"

_nutrients, 2025, doi:10.3390/nu17020216_

Round 1
Reviewer 1 Report
Comments and Suggestions for Authors
- The abstract is very long and gives too much information. It should be made more concise. I would advise a 200-250 word abstract, with some background and introduction to what is covered in the review.
- The very first sentence of the article, 'Chronic heart failure (CHF) patients face with formidable challenge of maintaining cardiac function.' is even not written correctly, unfortunately, which puts me off reading much further.
- There are also a lot of typos, which implies sloppiness on behalf of the author. E.g. Lines 65-66 'Although the pathologies of obesity, hypertension, aging and diabetes are associated with kidney function, the role of kidney in HFpEF is not suhggested.' Please thoroughly edit the article to correct typos, grammar and sentence syntax.
- For Figure 3, I would not put whole text citations in the figure, but just numbers in the figure. Then the rest of the figure could be spaced out.
- Otherwise, I do commend the use of figures to make schematics of what are some rather complex and inter-connected pathways. I think that's good.
- I also commend the use of a review article to bring the reader to a new hypothesis that could then be tested experimentally. It's unusual, but interesting.
Comments on the Quality of English LanguageAs commented above, the sentence structure is poor throughout, which limits my understanding of the content and limits my ability to fully review the article. It is not ready for publication as it is.
Author Response
Response to Reviewer 1:
- The abstract is very long and gives too much information. It should be made more concise. I would advise a 200-250 word abstract, with some background and introduction to what is covered in the review.
Response: The abstract is reduced to 245 words.
- The very first sentence of the article, 'Chronic heart failure (CHF) patients face with formidable challenge of maintaining cardiac function.' is even not written correctly, unfortunately, which puts me off reading much further.
Response: Corrected.
- There are also a lot of typos, which implies sloppiness on behalf of the author. E.g. Lines 65-66 'Although the pathologies of obesity, hypertension, aging and diabetes are associated with kidney function, the role of kidney in HFpEF is not suhggested.' Please thoroughly edit the article to correct typos, grammar and sentence syntax.
Response: Corrected.
- For Figure 3, I would not put whole text citations in the figure, but just numbers in the figure. Then the rest of the figure could be spaced out.
Response: We concur with the reviewer and modified some part of references in the figure. However, to inform the reader, it is important to cite the reference in fig that flows the chemical equation and mechanism.
- Otherwise, I do commend the use of figures to make schematics of what are some rather complex and inter-connected pathways. I think that's good.
Response: Corrected. We thank the reviewer for positive comments.
- I also commend the use of a review article to bring the reader to a new hypothesis that could then be tested experimentally. It's unusual, but interesting.
Response: It is better to keep hypothesis at the end of review, so that future studies can be directed.
Comments on the Quality of English Language
English is corrected throughout the review for a flow and readability.
As commented above, the sentence structure is poor throughout, which limits my understanding of the content and limits my ability to fully review the article. It is not ready for publication as it is.
Response: We have corrected most of the commented above, the sentences’ structure is improved throughout, which improves the understanding of the content and limitation to fully understandable. Now the article is easily readable.
Reviewer 2 Report
Comments and Suggestions for Authors
This review looks at the interplay between high fat diet , 1 carbon metabolism and HF.
In general, I found the first part of the review to be somewhat difficult to follow and not as clearly written. From page 5 onwards, the review was quite well written. This first section requires editing for grammar and english.
For example, in the abstract line 9 starting with "Recent studies....than HFrEF" should be rewritten. Line 67- "suhggested"- spelling etc...
The figures are quite crowded and would benefit from presented in a clearer fashion.
Line 81, the mutation of DNMT is a gain of function mutation in this case. For loss of function mutations/knockouts, the animals die early in life with decreased methylation. In the reference cited here, there is hypermethylation. This point should be clarified.
Lines 77-84 were not focussed and should be rewritten.
Comments on the Quality of English LanguageThe first 5 pages should be checked for English. The 2nd half was well written.
Author Response
Response to reviewer #2
In general, I found the first part of the review to be somewhat difficult to follow and not as clearly written. From page 5 onwards, the review was quite well written. This first section requires editing for grammar and english.
Response: Thank you for the positive comments on our manuscript. We have considered all the suggestion of the reviewer and revised according to the reviewer recommendation.
For example, in the abstract line 9 starting with "Recent studies....than HFrEF" should be rewritten. Line 67- "suhggested"- spelling etc...
Response: Re-written and corrected.
The figures are quite crowded and would benefit from presented in a clearer fashion.
Response: Figures are much clearer now.
Line 81, the mutation of DNMT is a gain of function mutation in this case. For loss of function mutations/knockouts, the animals die early in life with decreased methylation. In the reference cited here, there is hypermethylation. This point should be clarified.
Response: corrected.
Lines 77-84 were not focussed and should be rewritten.
Response: Now corrected and focused to the goal of the review article.
Comments on the Quality of English Language
The first 5 pages should be checked for English. The 2nd half was well written.
Response: Corrected.
Reviewer 3 Report
Comments and Suggestions for Authors
This is a review on high fat diet induces epigenetic 1-carbon metabolism, homo-2 cystinuria in renal-dependent HFpEF. Abstract: The content is easy to understand. Main text: I think the content is easy for readers to understand. Other: The method of extracting the literature this time is not clear. Therefore, please note this as a limitation of this paper.
Author Response
Response to reviewer #3
This is a review on high fat diet induces epigenetic 1-carbon metabolism, homo-2 cystinuria in renal-dependent HFpEF. Abstract: The content is easy to understand. Main text: I think the content is easy for readers to understand. Other: The method of extracting the literature this time is not clear. Therefore, please note this as a limitation of this paper.
Response: We thank the review for positive comments on our article.
Round 2
Reviewer 2 Report
Comments and Suggestions for Authors
My previous concerns have been addressed.
Author Response
We thank the editor for their critical comments, and we revised the manuscript by Editors recommendation as follows:
We started introduction by high fat diet induced HFpEF.
The abstract is 245 words.
The figures are discussed in detail.